# Fast Tucker Rank Reduction for Non-Negative Tensors Using Mean-Field Approximation

**Kazu Ghalamkari**[1,2]    **Mahito Sugiyama**[1,2]
[1]National Institute of Informatics
[2]The Graduate University for Advanced Studies, SOKENDAI
{gkazu,mahito}@nii.ac.jp

## Abstract

We present an efficient low-rank approximation algorithm for non-negative tensors. The algorithm is derived from our two findings: First, we show that rank-1 approximation for tensors can be viewed as a *mean-field approximation* by treating each tensor as a probability distribution. Second, we theoretically provide a sufficient condition for distribution parameters to reduce Tucker ranks of tensors; interestingly, this sufficient condition can be achieved by iterative application of the mean-field approximation. Since the mean-field approximation is always given as a closed formula, our findings lead to a fast low-rank approximation algorithm without using a gradient method. We empirically demonstrate that our algorithm is faster than the existing non-negative Tucker rank reduction methods and achieves competitive or better approximation of given tensors.

## 1 Introduction

A multidimensional array, or *tensor*, is a fundamental data structure in machine learning and statistical data analysis, and extraction of the essential information contained in tensors has been studied extensively [14, 20]. For second-order tensors – that is, matrices – *low-rank approximation* by singular value decomposition (SVD) is well established [13]. SVD always provides the best low-rank approximation in the sense of arbitrary unitarily invariant norms [30]. In contrast, the problem of low-rank approximation becomes much more challenging for tensors higher than the second order, where the question of how to define the rank of tensors is even nontrivial. To date, various types of ranks – the CP-rank [17, 24], the Tucker rank [9, 39], and the tubal rank [29] – have been proposed, and low-rank approximation of tensors in terms of one of the above two ranks has been widely studied. Furthermore, non-negative low-rank approximation has also been developed, not only for matrices such as NMF [26], but also for tensors [27]. In particular, non-negative Tucker decomposition (NTD) [21] and its efficient variant lraSNTD [43] approximate a given non-negative tensor by a tensor with the lower Tucker rank.

While these approximations have been widely used in various domains such as image classification [23], recommendation [37], and denoising [12], efficient low-rank approximation remains fundamentally challenging. Even the simplest case, the rank-1 approximation in terms of minimizing the Least Squares (LS) error between a given tensor and a low-rank tensor, is known to be NP-hard [16]. Various methods have been developed to efficiently find approximate solutions in polynomial runtime [7, 8, 10, 25, 42]. If we use the Kullback–Leibler (KL) divergence instead of the LS error as a cost function, we can alleviate the problem as the best rank-1 approximation can be obtained in the closed formula [19]. However, the general case of low-rank approximation in terms of the KL divergence is also still under development.

In this paper, we present a fast low-Tucker-rank approximation method for non-negative tensors. To date, the majority of low-rank approximation methods are based on gradient decent using the

35th Conference on Neural Information Processing Systems (NeurIPS 2021).

derivative of the cost function, which often requires careful tuning of initialization and/or a tolerance threshold. In contrast, our method is not based on a gradient method; the solution is directly obtained based on a closed formula, which we derive from information geometric treatment of tensors. Through an alternative parameterization of tensors by treating them as probability distributions in a statistical manifold, we theoretically provide a sufficient condition for such parameters, called the *bingo rule*, to reduce Tucker ranks of tensors. We then show that low-rank approximation is achieved by *m-projection*; this is one of the two canonical projections in information geometry [2], where a distribution (corresponding to a given non-negative tensor) is projected onto the subspace restricted by the bingo rule (corresponding to the set of non-negative low-rank tensors).

The key insight is that rank-1 approximation for non-negative tensors can be exactly solved by a *mean-field approximation*, a well-established method in physics that approximates a joint distribution by independent distributions [40], as we can represent any non-negative rank-1 tensor by a product of independent distributions. Moreover, we show that the bingo rule, our sufficient condition for tensor Tucker rank reduction, can be achieved by iterative applications of the mean-field approximation. This, combined with the fact that mean-field approximation is computed by $m$-projection in the closed form, enables us to derive our fast low-Tucker-rank approximation method without using a gradient method. Our theoretical analysis has a close relationship to [36], whose proposal, called Legendre decomposition, also uses information geometric parameterization of tensors and solves the problem of tensor decomposition by a projection onto a subspace. Although we use the same information geometric formulation of tensors, they did not provide any connection to the Tucker ranks, and Tucker rank reduction is not guaranteed by their approach. A limitation of our method is that it only treats non-negative tensors and cannot handle tensors with zero or negative values. Although experimental results show the usefulness of our method, even for tensors including zeros, we always assume that every element of an input tensor is strictly positive in our theoretical discussion.

## 2 The Proposed Rank Reduction Algorithm

We propose a Tucker rank reduction algorithm, called the *Legendre Tucker rank reduction* (LTR), which transforms a given non-negative tensor $\mathcal{P} \in \mathbb{R}_{\geq 0}^{I_1 \times \cdots \times I_d}$ into a tensor that approximates $\mathcal{P}$ with arbitrary reduced Tucker rank $(r_1, \ldots, r_d)$ specified by the user. The term "Legendre" of our algorithm comes from the fact that the theoretical support of our algorithm uses parameterization of tensors based on the *Legendre transformation*, which will be clarified in Section 3.

### 2.1 Problem setup and notation

First we define the Tucker rank of tensors and formulate the problem of non-negative Tucker rank reduction. The *Tucker rank* of a $d$th-order tensor $\mathcal{P} \in \mathbb{R}^{I_1 \times \cdots \times I_d}$ is defined as a tuple $(\mathrm{Rank}(\mathcal{P}^{(1)}),$ $\ldots, \mathrm{Rank}(\mathcal{P}^{(d)}))$, where each $\mathcal{P}^{(k)} \in \mathbb{R}^{I_k \times \prod_{m \neq k} I_m}$ is the mode-$k$ expansion of the tensor $\mathcal{P}$ and $\mathrm{Rank}(\mathcal{P}^{(k)})$ denotes the matrix rank of $\mathcal{P}^{(k)}$ [11, 15, 39]. See Supplement for definition of the mode-$k$ expansion. If the Tucker rank of a tensor $\mathcal{P}$ is $(r_1, \ldots, r_d)$, it can be always decomposed as

$$\mathcal{P} = \sum_{i_1=1}^{r_1} \cdots \sum_{i_d=1}^{r_d} \mathcal{G}_{i_1,\ldots,i_d} \boldsymbol{a}_{i_1}^{(1)} \otimes \boldsymbol{a}_{i_2}^{(2)} \otimes \cdots \otimes \boldsymbol{a}_{i_d}^{(d)}$$

with a tensor $\mathcal{G} \in \mathbb{R}^{r_1 \times \cdots \times r_d}$, called the core tensor of $\mathcal{P}$, and vectors $\boldsymbol{a}_{i_k}^{(k)} \in \mathbb{R}^{I_k}$, $i_k \in \{1, \ldots, r_k\}$, for each $k \in \{1, \ldots, d\}$, where $\otimes$ denotes the Kronecker product [24].

If every element of the Tucker rank is the same as each other element, it coincides with the CP rank. In this paper, we say that a tensor is rank-1 if its Tucker rank is $(1, \ldots, 1)$. The problem with non-negative Tucker rank reduction is approximating a given non-negative tensor by a non-negative lower Tucker rank tensor. We denote by $[d] = \{1, 2, \ldots, d\}$ for a positive integer $d$ and denote by $\mathcal{P}_{a^{(k)}:b^{(k)}}$ the subtensor obtained by fixing the range of $k$th index to only from $a$ to $b$.

### 2.2 The LTR Algorithm

In LTR, we use the rank-1 approximation method that always finds the rank-1 tensor that minimizes the KL divergence from an input tensor [19]. The optimal rank-1 tensor $\overline{\mathcal{P}}$ of $\mathcal{P}$ is given by

$$\overline{\mathcal{P}} = \lambda \boldsymbol{s}^{(1)} \otimes \boldsymbol{s}^{(2)} \otimes \cdots \otimes \boldsymbol{s}^{(d)}, \tag{1}$$

**Algorithm 1:**

**input** : Tensor $\mathcal{P}$, target Tucker rank $\mathbf{r} = (r_1, \ldots, r_d)$
**output**: Rank reduced tensor $\mathcal{Q}$
LTR($\mathcal{P}$,**r**)
  $(I_1, \ldots, I_d) \leftarrow$ the size of the input tensor $\mathcal{P}$
  **foreach** $k = 1, \ldots, d$ **do**
    Construct $\{c_1, \ldots, c_{r_k}\} \subseteq [I_k]$ by random sampling from $[I_k]$ without replacement,
      where we always assume that $c_1 = 1$ and $c_i < c_{i+1}$.
    **foreach** $l = 1, \ldots, r_k$ **do**
      **if** $c_l \neq c_{l+1} - 1$ **then**
        Replace the subtensor $\mathcal{P}_{c_l^{(k)}:c_{l+1}^{(k)}-1}$ of $\mathcal{P}$ by its rank-1 approximation as
        $\mathcal{P}_{c_l^{(k)}:c_{l+1}^{(k)}-1} \leftarrow \text{BESTRANK1}(\mathcal{P}_{c_l^{(k)}:c_{l+1}^{(k)}-1})$

  $\mathcal{Q} \leftarrow \mathcal{P}$
  **return** $\mathcal{Q}$
BESTRANK1($\mathcal{P}$)
  $(I_1, \ldots, I_d) \leftarrow$ the size of the input tensor $\mathcal{P}$
  **foreach** $k = 1, \ldots, d$ **do**
    **foreach** $i_k = 1, \ldots, I_k$ **do**
      $s_{i_k}^{(k)} \leftarrow \sum_{i_1=1}^{I_1} \cdots \sum_{i_{k-1}=1}^{I_{k-1}} \sum_{i_{k+1}=1}^{I_{k+1}} \cdots \sum_{i_d=1}^{I_d} \mathcal{P}_{i_1, \ldots i_{k-1}, i_k, i_{k+1}, \ldots, i_d}$

  $\lambda \leftarrow$ sum of all elements of $\mathcal{P}$
  $\overline{\mathcal{P}} \leftarrow \lambda \boldsymbol{s}^{(1)} \otimes \boldsymbol{s}^{(2)} \otimes \cdots \otimes \boldsymbol{s}^{(d)}$
  **return** $\overline{\mathcal{P}}$

where each $\boldsymbol{s}^{(k)} = (s_1^{(k)}, \ldots, s_{I_k}^{(k)})$ with $k \in [d]$ is defined as

$$s_{i_k}^{(k)} = \sum_{i_1=1}^{I_1} \cdots \sum_{i_{k-1}=1}^{I_{k-1}} \sum_{i_{k+1}=1}^{I_{k+1}} \cdots \sum_{i_d=1}^{I_d} \mathcal{P}_{i_1, \ldots, i_d}$$

and $\lambda$ is the $(1-d)$-th power of the sum of all elements of $\mathcal{P}$.

Now we introduce LTR, which iteratively applies the above rank-1 approximation to subtensors of a tensor $\mathcal{P} \in \mathbb{R}^{I_1 \times \cdots \times I_d}$. When we reduce the Tucker rank of $\mathcal{P}$ to $(r_1, \ldots, r_d)$, LTR performs the following two steps for each $k \in [d]$:

**Step 1:** We construct $C = \{c_1, \ldots, c_{r_k}\} \subseteq [I_k]$ by random sampling from $[I_k]$ without replacement, where we always assume that $c_1 = 1$ and $c_l < c_{l+1}$ for every $l \in [r_k - 1]$.

**Step 2:** For each $l \in [r_k]$, if $c_l \neq c_{l+1} - 1$ holds, we replace the subtensor $\mathcal{P}_{c_l^{(k)}:c_{l+1}^{(k)}-1}$ of $\mathcal{P}$ by its rank-1 approximation obtained by Equation (1).

Step 1 of LTR requires $O(r_1 + r_2 + \cdots + r_d)$ since we only need to sample integers from $1, 2, \ldots, I_k$ for each $k \in [d]$ using the Fisher-Yates method. Since the above procedure repeats the best rank-1 approximation at most $r_1 r_2 \ldots r_d$ times, the worst computational complexity of LTR is $O(r_1 r_2 \ldots r_d I_1 I_2 \ldots I_d)$. The choice of $C$ in **Step 1** is arbitrary, which means that another strategy can be used. For example, if we know that some parts of an input tensor are less important than others, we can directly choose these indices for $C$ instead of random sampling to obtain a more accurate reconstructed tensor.

We provide the algorithm of LTR in algorithmic format in Algorithm 1.

## 3 Theoretical Analysis of LTR

To theoretically guarantee that LTR always reduces the Tucker rank to $(r_1, \ldots, r_d)$, in the following subsections, we introduce information geometric analysis of the low-rank tensor approximation by treating each tensor as a probability distribution. The proof of the propositions are given in the supplementary material.

## 3.1 Modeling tensors as probability distributions

Our key idea for the derivation of LTR is the information geometric treatment of positive tensors in a probability space. We use the special case of a log-linear model on partially ordered sets (poset) introduced by Sugiyama et al. [35] for our probabilistic formulation as they have studied tensors on a statistical manifold [36] using the formulation. We see that the low Tucker rank tensor approximation for positive tensors in terms of the KL divergence is formulated as a convex optimization problem.

In the following, we assume that an input positive tensor $\mathcal{P} \in \mathbb{R}_{>0}^{I_1 \times \cdots \times I_d}$ is always normalized so that the sum of all elements is 1, and we regard $\mathcal{P}$ as a discrete probability distribution whose sample space is the index set of tensors $\Omega_d = [I_1] \times \cdots \times [I_d]$. Any positive normalized tensor can be described by canonical parameters $(\theta)_{i_1,\ldots,i_d} = (\theta_{1,\ldots,1}, \ldots, \theta_{I_1,\ldots,I_d})$ as

$$\mathcal{P}_{i_1,\ldots,i_d} = \exp\left( \sum_{i'_1=1}^{i_1} \cdots \sum_{i'_d=1}^{i_d} \theta_{i'_1,\ldots,i'_d} \right). \tag{2}$$

The condition of normalization is exposed on $\theta_{1,\ldots,1}$ with $\Omega_d^+ = \Omega_d \backslash \{(1,\ldots,1)\}$ as

$$\theta_{1,\ldots,1} = -\log\left( \sum_{(i_1,\ldots,i_d) \in \Omega_d^+} \exp\left( \sum_{i'_1=1}^{i_1} \cdots \sum_{i'_d=1}^{i_d} \theta_{i'_1,\ldots,i'_d} \right) \right). \tag{3}$$

This belongs to the log-linear model, and a parameter vector $(\theta)_{i_1,\ldots,i_d}$ in Equation (2) uniquely identifies the normalized positive tensor $\mathcal{P}$. Therefore, $(\theta)_{i_1,\ldots,i_d}$ can be used as an alternative representation of $\mathcal{P}$. If we see each probabilistic distribution – a normalized tensor in our case – as a point in a statistical manifold, $(\theta)_{i_1,\ldots,i_d}$ corresponds to a coordinate of the manifold, which is a typical approach in information geometry and known as the $\theta$-coordinate system [2].

It is known that any distribution in an exponential family $\log p_\theta(x) = C(x) + \sum_{i=1}^{k} \theta_i F_i(x) - \psi(\theta)$ for the normalization factor $\psi(\theta)$ and canonical parameters $(\theta)_i$ can be also uniquely identified by expectation parameters $\eta_i = \mathbb{E}_{p_\theta}[F_i]$ [4]. In our modeling in Equation (2), which clearly belongs to the exponential family, each value of the vector of $\eta$-parameters $(\eta)_{i_1,\ldots,i_d}$ is written as follows and uniquely identifies a normalized positive tensor $\mathcal{P}$:

$$\eta_{i_1,\ldots,i_d} = \sum_{i'_1=i_1}^{I_1} \cdots \sum_{i'_d=i_d}^{I_d} \mathcal{P}_{i'_1,\ldots,i'_d}. \tag{4}$$

Hence we can also use $(\eta)_{i_1,\ldots,i_d}$ as a coordinate system of the set of distributions, known as the $\eta$-coordinate system in information geometry [2]. As shown in Sugiyama et al. [35], by using the Möbius function [32, 34] inductively defined as

$$\mu_{i_1,\ldots,i_d}^{i'_1,\ldots,i'_d} = \begin{cases} 1 & \text{if } i_k = i'_k \text{ for all } k \in [d], \\ -\prod_{k=1}^{d} \sum_{j_k=i_k}^{i'_k-1} \mu_{i_1,\ldots,i_d}^{j_1,\ldots,j_d} & \text{if } i_k < i'_k \text{ for all } k \in [d], \\ 0 & \text{otherwise,} \end{cases}$$

each distribution $\mathcal{P}$ can be described as

$$\mathcal{P}_{i_1,\ldots,i_d} = \sum_{(i'_1,\ldots,i'_d) \in \Omega_d} \mu_{i_1,\ldots,i_d}^{i'_1,\ldots,i'_d} \eta_{i'_1,\ldots,i'_d}. \tag{5}$$

using the $\eta$-coordinate system. The normalization condition is realized as $\eta_{1,\ldots,1} = 1$.

In information geometry, it is known that $\theta$- and $\eta$-coordinates are connected via Legendre transformation [2]. The remarkable property of this pair of coordinate systems is that they are orthogonal with each other and we can combine them to define the $(\theta, \eta)$-coordinate as a mixture coordinate. If we specify *either* $\theta_{i_1,\ldots,i_d}$ *or* $\eta_{i_1,\ldots,i_d}$ for every $(i_1,\ldots,i_d) \in \Omega_d$, we can always uniquely identify a normalized positive tensor $\mathcal{P}$.

## 3.2 Bingo rule: $\theta$ representation of Tucker rank condition

We theoretically derive the relationship between the Tucker rank of tensors and conditions on the $\theta$-coordinate system. We achieve Tucker rank reduction using such conditions instead of directly imposing constraints on elements of tensors and solving it using classical methods such as the Lagrange multipliers. The advantage of our approach is that we can formulate low-rank approximation as a projection of a distribution $\mathcal{P}$ on a subspace of the $(\theta, \eta)$-space, which always becomes a convex optimization due to the flatness of its subspace [2].

**Definition 1** (Bingo). *Let* $(\theta)_{ij}^{(k)} = (\theta_{11}^{(k)}, \ldots, \theta_{I_k K}^{(k)})$ *with* $K = \prod_{m \neq k} I_m$ *be the $\theta$-coordinate representation of the mode-$k$ expansion of a tensor* $\mathcal{P} \in \mathbb{R}_{>0}^{I_1 \times \cdots \times I_d}$. *If there exists an integer* $i \in [I_k] \setminus \{1\}$ *such that* $\theta_{ij}^{(k)} = 0$ *for all* $j \in [K] \setminus \{1\}$, *we say that* $\mathcal{P}$ *has a* bingo *on mode-$k$.*

**Proposition 1** (Bingo and Tucker rank). *If there are $b_k$ bingos on mode-$k$, it holds that*

$$\mathrm{Rank}(\mathcal{P}^{(k)}) \leq I_k - b_k.$$

Therefore, for any tensor $\mathcal{P} \in \mathbb{R}_{>0}^{I_1 \times \cdots \times I_d}$ such that it has $b_k$ bingos for each $k$th index, we can always guarantee that its Tucker rank is at most $(I_1 - b_1, \ldots, I_d - b_d)$.

### 3.3 Projection theory for LTR

We achieve low Tucker-rank approximation by projection onto a submanifold; that is, our proposed algorithm TLR performs projection in the viewpoint of information geometry. In a statistical manifold parameterized by $\theta$- and $\eta$-coordinate systems, two types of geodesics, $m$- and $e$-geodesics from a point $\mathcal{P}$ to $\mathcal{Q}$ in a manifold can be defined as

$$\{ \mathcal{R}_t \mid \mathcal{R}_t = (1-t)\mathcal{P} + t\mathcal{Q} \}, \quad \{ \mathcal{R}_t \mid \log \mathcal{R}_t = (1-t)\log \mathcal{P} + t \log \mathcal{Q} - \phi(t) \},$$

respectively, where $0 \leq t \leq 1$ and $\phi(t)$ is a normalization factor to keep $\mathcal{R}_t$ to be a distribution. In the above definition, we regard $\mathcal{P}$ and $\mathcal{Q}$ as its corresponding coordinates points $(\theta)_{i_1,\ldots,i_d}$ or $(\eta)_{i_1,\ldots,i_d}$. A subspace is called $e$-flat when any $e$-geodesic connecting any two points in a subspace is included in the subspace. The vertical descent of an $m$-geodesic from a point $\mathcal{P}$ to $\mathcal{Q}$ in an $e$-flat subspace $M_e$ is called $m$-projection. Similarly, $e$-projection is obtained when we replace all $e$ with $m$ and $m$ with $e$. The flatness of subspaces guarantees the uniqueness of the projection destination. The projection destination $\mathcal{R}_m$ or $\mathcal{R}_e$ obtained by $m$- or $e$-projection onto $M_e$ or $M_m$ minimizes the following KL divergence,

$$\mathcal{R}_m = \underset{\mathcal{Q} \in M_e}{\mathrm{argmin}} \, D_{\mathrm{KL}}(\mathcal{P}; \mathcal{Q}), \quad \mathcal{R}_e = \underset{\mathcal{Q} \in M_m}{\mathrm{argmin}} \, D_{\mathrm{KL}}(\mathcal{Q}; \mathcal{P}).$$

LTR conducts an $m$-projection from an input positive tensor $\mathcal{P}$ onto the low-rank space $\mathcal{B}$, which is the set of tensors, each of which satisfies a given set of bingos. In practical situations, a Tucker rank constraint $(r_1, \ldots, r_d)$ is given as an input parameter, and LTR first implicitly constructs $\mathcal{B}$ in **Step 1** by translating the Tucker rank condition into bingos, where any tensor in $\mathcal{B}$ has the Tucker rank at most $(r_1, \ldots, r_d)$. We call the low-rank subspace $\mathcal{B}$ bingo space. Since the low-rank space $\mathcal{B}$ is $e$-flat, the $m$-projection that minimizes the KL divergence is uniquely determined.

In $m$-projection into a space where some $\theta$-parameters are constrained to be 0, every $\eta$-parameter with the same index with some restricted $\theta$-parameter does not change [2]. Therefore, we always have

$$\eta_{i_1,\ldots,i_d} = \eta'_{i_1,\ldots,i_d} \text{ if } \theta'_{i_1,\ldots,i_d} \neq 0, \tag{6}$$

where $\eta_{i_1,\ldots,i_d}$ is the $\eta$-coordinate of an input positive tensor $\mathcal{P}$ and $\eta'_{i_1,\ldots,i_d}$ is that of the destination of $m$-projection onto $\mathcal{B}$ from $\mathcal{P}$. Using the conservation law for the $\eta$-coordinate is our key insight to conduct the efficient $m$-projection onto the bingo space $\mathcal{B}$.

**Relationship to Legendre decomposition** Legendre decomposition [36] is a tensor decomposition method based on an information geometric view. However, their concept differs from ours in the following aspect. In the Legendre decomposition, any single point in a subspace that has some constraints on the $\theta$-coordinate is taken as the initial state and moves by gradient descent inside the subspace to minimize the KL divergence from an input tensor. This operation is an $e$-projection, where the constrained $\theta$-coordinates do not change from the initial state. In contrast, we employ the $m$-projection from the input tensor onto the low-rank space by fixing some $\eta$-coordinates, as shown in Equation (6). Using this conservation law for $\eta$-coordinates, we can obtain an exact analytical representation of the coordinates of the projection destination without using a gradient method. Figure 1(a) illustrates the relationship between our approach and Legendre decomposition. Moreover, the Tucker rank is not discussed in the Legendre decomposition, so it is not guaranteed that Legendre decomposition reduces the Tucker rank, which is in contrast to our method.

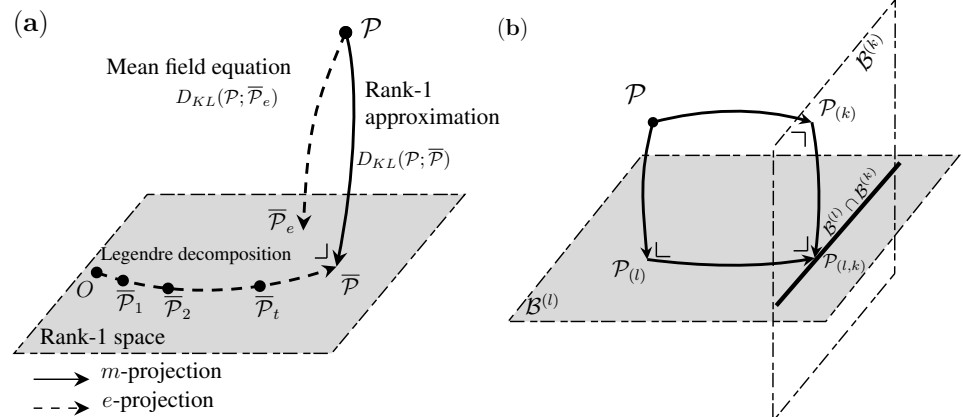

Figure 1: (a) The relationship among rank-1 approximation, Legendre decomposition [36], and mean-field equation, where we assume that the same bingo space is used in Legendre decomposition. A solid line illustrates $m$-projection with fixing one-body $\eta$ parameters. $\mathcal{P}$ is an input positive tensor and $\overline{\mathcal{P}}$ is the rank-1 tensor that minimizes the KL divergence from $\mathcal{P}$. $\mathcal{O}$ is an initial point of Legendre decomposition, which is usually a uniform distribution. $\overline{\mathcal{P}}_t$ is a tensor of the $t$-th step of gradient descent in Legendre Decomposition. (b) The $m$-projection of a common space of two different bingo spaces from $\mathcal{P}$ can be achieved by $m$-projection into one bingo space and then $m$-projecting into the other bingo space.

### 3.4 Rank-1 approximation as mean-field approximation

Here we focus on the problem of rank-1 approximation for positive tensors and show the fundamental relationship with the *mean-field theory*. In the following, we refer to the subspace consisting of positive rank-1 tensors as a *rank-1 space*. We use the overline for rank-1 tensors; that is, $\overline{\mathcal{P}}$ is a rank-1 tensor and $\overline{\theta}, \overline{\eta}$ are corresponding parameters of $\theta$- and $\eta$-coordinates. Let a *one-body* parameter be a parameter of which at least $d-1$ indices are 1; for example, $\theta_{1,1,3,1}$ and $\eta_{1,5,1,1}$ are one-body parameters for $d = 4$. Parameters other than one-body parameters are called *many-body* parameters. These namings come from the Boltzmann machine [1], which is a special case of the log-linear model [35], where a one-body parameter corresponds to a bias and a many-body parameter to a weight. We also use the following notation for one-body parameters of a $d$th-order tensor,

$$\theta_j^{(k)} \equiv \theta_{\underbrace{1,\ldots,1}_{k-1},j,\underbrace{1,\ldots,1}_{d-k}}, \quad \eta_j^{(k)} \equiv \eta_{\underbrace{1,\ldots,1}_{k-1},j,\underbrace{1,\ldots,1}_{d-k}} \quad \text{for each } k \in [d].$$

The rank-1 condition for positive tensors is described as follows using many-body $\theta$ parameters as a special case of Proposition 1.

**Proposition 2** (rank-1 condition on $\theta$). *For any positive tensor $\overline{\mathcal{P}}$, $\operatorname{rank}(\overline{\mathcal{P}}) = 1$ if and only if all of its many-body $\overline{\theta}$ parameters are 0.*

Note that the bingo constraints on $\theta$-coordinates become unique if the target Tucker rank is $(1, \ldots, 1)$, hence the projection always finds the best rank-1 tensor that minimizes the KL divergence from an input tensor. This means that our information geometric formulation of the rank-1 approximation leads to the same result with the closed formula of the best rank-1 approximation given by Huang and Sidiropoulos [19]. Indeed, using the factorizability of $\eta$-parameters for rank-1 tensors, we can reproduce the closed formula in [19] (see Supplement for its proof).

We consider a rank-1 positive tensor $\overline{\mathcal{P}} \in \mathbb{R}_{>0}^{I_1 \times \cdots \times I_d}$ and show that it is represented as a product of independent distributions, which leads to an analogy with the mean-field theory. In the rank-1 space, the normalization condition for $d$th-order tensors imposed on $\theta$ parameters is given as

$$\overline{\theta}_{1,\ldots,1} = -\log \prod_{k=1}^{d} \left( 1 + \sum_{i_k=2}^{I_k} \exp \left( \sum_{i'_k=2}^{i_k} \overline{\theta}_{i'_k}^{(k)} \right) \right) \tag{7}$$

by assigning 0 to every many-body $\theta$ parameter in Equation (3). Note that the empty sum is treated as zero. Next, by substituting 0 for all many-body parameters in our model in Equation (2), we obtain

$$\overline{\mathcal{P}}_{j_1,\ldots,j_d} = \exp\left(\overline{\theta}_{1,\ldots,1}\right) \prod_{k=1}^{d} \exp\left(\sum_{j'_k=2}^{j_k} \overline{\theta}_{j'_k}^{(k)}\right) \overset{(7)}{=} \prod_{k=1}^{d} \frac{\exp\left(\sum_{j'_k=2}^{j_k} \overline{\theta}_{j'_k}^{(k)}\right)}{1 + \sum_{i_k=2}^{I_k} \exp\left(\sum_{i'_k=2}^{i_k} \overline{\theta}_{i'_k}^{(k)}\right)} \equiv \prod_{k=1}^{d} s_{j_k}^{(k)},$$

where $\boldsymbol{s}^{(k)} \in \mathbb{R}^{I_k}$ is a positive first-order tensor normalized as $\sum_{j_k=1}^{I_k} s_{j_k}^{(k)} = 1$. Since exactly one element is used in each $\boldsymbol{s}^{(k)}$ to determine $\mathcal{P}_{j_1,\ldots,j_d}$, we can regard $\boldsymbol{s}^{(k)}$ as a probability distribution with a single random variable $j_k \in [I_k]$. The above discussion means that any positive rank-1 tensor can be represented as a product of normalized independent distributions.

The operation of approximating a joint distribution as a product of independent distributions is called *mean-field approximation*. The mean-field approximation was invented in physics for discussing phase transition in ferromagnets [40]. Nowadays, it appears in a wide range of domains such as statistics [31], game theory [6, 28], and information theory [5]. From the viewpoint of information geometry, Tanaka [38] developed a theory of mean-field approximation for Boltzmann machines [1], which is defined as $p(\boldsymbol{x}) = \exp(\sum_i b_i x_i + \sum_{ij} w_{ij} x_i x_j)$ for a binary random variable vector $\boldsymbol{x} \in \{0, 1\}^n$ with a bias parameter $\boldsymbol{b} = (b)_i \in \mathbb{R}^n$ and an interaction parameter $\boldsymbol{W} = (w_{ij}) \in \mathbb{R}^{n \times n}$. To illustrate that a rank-1 approximation can be regarded as a mean-field approximation, we prepare the mean-field theory of Boltzmann machines, as follows.

The mean-field approximation of Boltzmann machines is formulated as the projection from a given distribution onto the $e$-flat subspace consisting of distributions whose interaction parameters $w_{ij} = 0$ for all $i$ and $j$, which is called a factorizable subspace. Since the distribution with the constraint $w_{ij} = 0$ for all $i$ and $j$ can be decomposed into a product of independent distributions, we can approximate a given distribution as a product of independent distribution by the projection onto a factorizable subspace. The $m$-projection onto the factorizable subspace requires knowing the expectation value $\eta_i \equiv \mathbb{E}[x_i]$ of an input distributions and requires $O(2^n)$ computational cost [3], so we usually approximate it by replacing the $m$-projection with $e$-projection. The $e$-projection is usually conducted by a self-consistent equation called *mean-field equation*.

The analogy of LTR and mean-field theory is clear. In our modeling, a joint distribution $\mathcal{P}$ is approximated by a product of independent distributions $\boldsymbol{s}^{(k)}$ by projecting $\mathcal{P}$ onto the subspace such that all many-body $\theta$ parameters are 0, leading to the rank-1 tensor $\overline{\mathcal{P}}$. Since we can compute expectation parameters $\eta$ by simply summing the input positive tensor in each axial direction, $m$-projection can be directly performed in our formulation, which is computationally infeasible in the case of Boltzmann machines due to $O(2^n)$ cost. Moreover, the rank-1 space has the same property as the factorizable subspace of Boltzmann machines; that is, $\eta$ can be easily computed from $\theta$ (see Supplement for more details). This is the first time that the relationship between mean-field approximation and the tensor rank reduction has been demonstrated. The sketch of the relationship among mean-field equations and the rank-1 approximation is illustrated in Figure 1(a).

### 3.5 Derivation of LTR

Finally, we prove that LTR successfully reduces the Tucker rank by extending the above discussion. We formulate Tucker rank reduction as an $m$-projection onto a specific bingo space. This bingo space is constructed in **Step 1** of LTR. Then, in **Step 2**, we can perform the $m$-projection using the closed formula of the rank-1 approximation without a gradient method. We first discuss the case in which the rank of only one mode is reduced, followed by discussing the case in which the ranks of two modes are reduced.

**When the rank of only one mode is reduced** Let us assume that the target Tucker rank is $(I_1, \ldots, I_{k-1}, r_k, I_{k+1}, \ldots, I_d)$ for an input positive tensor $\mathcal{P} \in \mathbb{R}_{>0}^{I_1 \times \cdots \times I_d}$. Let $\mathcal{B}^{(k)}$ be the set of tensors satisfying bingos and $\Omega_{\mathcal{B}^{(k)}}$ be the set of bingo indices for mode-$k$ constructed in **Step 1** of LTR:

$$\mathcal{B}^{(k)} = \{ \mathcal{P} \mid \theta_{i_1,\ldots,i_d} = 0 \text{ for } (i_1, \ldots, i_d) \in \Omega_{\mathcal{B}^{(k)}} \}. \tag{8}$$

Note that $\mathcal{P} \in \mathcal{B}^{(k)}$ implies that the Tucker rank of $\mathcal{P}$ is at most $(I_1, \ldots, I_{k-1}, r_k, I_{k+1}, \ldots, I_d)$. Let $\tilde{\mathcal{P}}$ be the destination of the $m$-projection and $\tilde{\theta}, \tilde{\eta}$ be its corresponding parameters of $\theta$- and

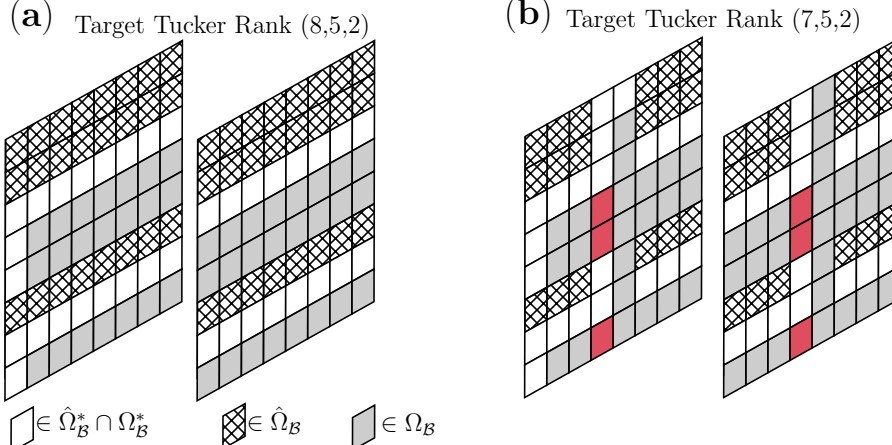

$\square \in \hat{\Omega}_{\mathcal{B}}^* \cap \Omega_{\mathcal{B}}^* \qquad \boxtimes \in \hat{\Omega}_{\mathcal{B}} \qquad \square \in \Omega_{\mathcal{B}}$

Figure 2: Examples of LTR for $(8, 8, 2)$ tensor. $\Omega_{\mathcal{B}}$ is bingo index. Tensor values and their $\eta$-parameters on $\hat{\Omega}_{\mathcal{B}}$ do not change, and the $\eta$-parameters on $\hat{\Omega}_{\mathcal{B}}^* \cap \Omega_{\mathcal{B}}^*$ also do not change. (a) Target rank is $(8, 5, 2)$. Bingos are on mode-2. In this example, the number of contiguous blocks is 2. (b) Target rank is $(7, 5, 2)$. Bingos are on modes 1 and 2 and the number of contiguous blocks is 1 and 2 for modes 1 and 2, respectively. We assume that we project a tensor onto $\mathcal{B}^{(1)}$, followed by projecting it onto $\mathcal{B}^{(2)}$. After the second $m$-projection, the $\theta$-parameters on red panels are overwritten. However, these values remain to be zero after the second $m$-projection in our LTR.

$\eta$-coordinates. From the definition of $m$-projection and the conservation low of $\eta$ in Equation (6), the tensor $\tilde{\mathcal{P}}$ should satisfy

$$\tilde{\theta}_{i_1,\ldots,i_d} = 0 \text{ for } (i_1,\ldots,i_d) \in \Omega_{\mathcal{B}^{(k)}}, \tag{9}$$

$$\tilde{\eta}_{i_1,\ldots,i_d} = \eta_{i_1,\ldots,i_d} \text{ for } (i_1,\ldots,i_d) \notin \Omega_{\mathcal{B}^{(k)}}. \tag{10}$$

As we can see from Equation (5), $\eta$-parameters identify a tensor $\mathcal{P}$. In particular, to identify each element $\mathcal{P}_{i_1,\ldots,i_d}$, we require $\eta$-parameters on only $(i_1,\ldots,i_d) \in \{i_1, i_1+1\} \times \{i_2, i_2+1\} \times \cdots \times \{i_d, i_d+1\}$. For example, if $d = 2$, $\mathcal{P}_{i_1,i_2} = \eta_{i_1+1,i_2+1} - \eta_{i_1+1,i_2} - \eta_{i_1,i_2+1} + \eta_{i_1,i_2}$ [35]. This leads to the fact that, for $\hat{\Omega}_{\mathcal{B}^{(k)}}$ such that

$$\hat{\Omega}_{\mathcal{B}^{(k)}} = \{ (i_1,\ldots,i_d) \mid \{i_1, i_1+1\} \times \cdots \times \{i_d, i_d+1\} \nsubseteq \Omega_{\mathcal{B}^{(k)}} \}, \tag{11}$$

it holds that $\mathcal{P}_{i_1,\ldots,i_d} = \tilde{\mathcal{P}}_{i_1,\ldots,i_d}$ for $(i_1,\ldots,i_d) \in \hat{\Omega}_{\mathcal{B}^{(k)}}$. Therefore, all we have to do to reduce the Tucker rank is to change values for only $(i_1,\ldots,i_d) \notin \hat{\Omega}_{\mathcal{B}^{(k)}}$. Such adjustable parts of $\mathcal{P}$ can be divided into some contiguous blocks, and we call each of them a subtensor of $\mathcal{P}$ on mode-$k$. In Figure 2, for example, we can find two subtensors $\mathcal{P}_{3(1):5(5)}$ and $\mathcal{P}_{7(1):8(1)}$. By conducting the rank-1 approximation introduced in Section 2 onto each subtensor, the rank-1 bingo condition ensures the $\theta$ condition in Equation (9) and the conservation low of $\eta$ in Equation (6) for the best rank-1 approximation ensures the $\eta$ condition in Equation (10). Therefore, we obtain the tensor $\tilde{\mathcal{P}}$ satisfying Equation (9) and (10) simultaneously by the rank-1 approximations on each subtensor on mode-$k$, which always belongs to $\mathcal{B}^{(k)}$.

**When the rank of only two modes are reduced** Let us assume that the target Tucker rank of mode-$k$ is $r_k$ and that of mode-$l$ is $r_l$. In this case, we need to consider two bingo spaces $\mathcal{B}^{(k)}$ and $\mathcal{B}^{(l)}$ associated with bingo index sets $\Omega_{\mathcal{B}^{(k)}}$ and $\Omega_{\mathcal{B}^{(l)}}$. Let $\mathcal{P}_{(k)}$ be the resulting tensor of $m$-projection of $\mathcal{P}$ onto $\mathcal{B}^{(k)}$ and $\mathcal{P}_{(k,l)}$ be the resulting tensor of $m$-projection of $\mathcal{P}_{(k)}$ onto $\mathcal{B}^{(l)}$. The parameters of $\mathcal{P}_{(k,l)}$ satisfy

$$\tilde{\theta}_{i_1,\ldots,i_d} = 0 \text{ if } (i_1,\ldots,i_d) \in \Omega_{\mathcal{B}^{(k)}} \cup \Omega_{\mathcal{B}^{(l)}}, \tag{12}$$

$$\tilde{\eta}_{i_1,\ldots,i_d} = \eta_{i_1,\ldots,i_d} \text{ otherwise.} \tag{13}$$

It is obvious that the $m$-projection from $\mathcal{P}_{(k)}$ onto $\mathcal{B}^{(l)}$ is equivalent to the $m$-projection from $\mathcal{P}$ onto $\mathcal{B} = \mathcal{B}^{(l)} \cap \mathcal{B}^{(k)}$ The conservation low of $\eta$-parameters ensures Equation (12) after the rank-1

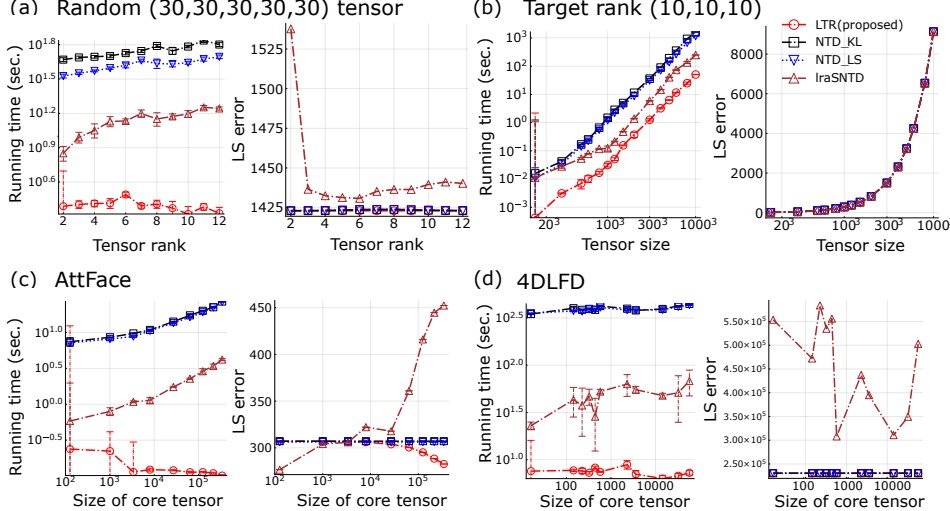

Figure 3: Experimental results for synthetic (a, b) and real-world (c, d) datasets. Mean errors $\pm$ standard error for 20 times iterations are plotted. (a) The horizontal axis is $r$ for target tensor rank $(r, r, r, r, r)$. (b) The horizontal axis is $n^3$ for input $(n, n, n)$ tensor. (c, d) The horizontal axis is the number of elements of the core tensor.

approximation of subtensors of $\mathcal{P}_{(k)}$ in the mode-$l$. In this operation, the part of $\theta$-parameters which are set to be 0 in the previous $m$-projection onto $\mathcal{B}^{(k)}$ from $\mathcal{P}$ seems to be overwritten (see red panels in Figure 2(b)). However, as shown in Proposition 6 in Supplement, when the rank-1 approximation of a tensor where some one-body $\theta$-parameters are already zero, the values of such one-body $\theta$-parameters after the rank-1 approximation remains to be zero. As a result, Equation (12) holds; that is, we can obtain the tensor $\mathcal{P}_{(k,l)}$ satisfying Equation (12) and (13) simultaneously by the rank-1 approximations on each subtensor of $\mathcal{P}_{(k)}$ on mode-$l$. We can also immediately confirm $\mathcal{P}_{(l,k)} = \mathcal{P}_{(k,l)}$; that is, the projection order does not matter. The projection sketch is shown in Figure 1(b).

Using the above discussion, we can derive **Step 2** for the general case of Tucker rank reduction.

**Theorem 1** (LTR). *For a positive tensor $\mathcal{P} \in \mathbb{R}_{>0}^{I_1 \times \cdots \times I_d}$, the $m$-projection destination onto the bingo space $\mathcal{B} = \mathcal{B}^{(1)} \cap \cdots \cap \mathcal{B}^{(d)}$ is given as an iterative application of $m$-projection $d$ times, starting from $\mathcal{P}$ onto subspace $\mathcal{B}^{(1)}$, then from there onto subspace $\mathcal{B}^{(2)}$, ..., and finally onto subspace $\mathcal{B}^{(d)}$.*

The result of LTR does not depend on the projection order. The conservation law of $\eta$-parameters during the $m$-projection also ensures that LTR conserves the sum in each axial direction of the input tensor (see Supplement for its proof).

Since the $m$-projection minimizes the KL divergence from the input onto the bingo space, LTR always provides the best low-rank approximation, in the specified bingo space $\mathcal{B}$, that is, for a given non-negative tensor $\mathcal{P}$, the output $\mathcal{Q}^*$ of LTR satisfies that

$$\mathcal{Q}^* = \underset{\mathcal{Q} \in \mathcal{B}}{\arg\min} \, D_{\mathrm{KL}}(\mathcal{P}; \mathcal{Q}).$$

The usual low-rank approximation without the bingo-space requirement approximates a tensor by a linear combination of appropriately chosen bases. In contrast, our method with the bingo-space requirement approximates a tensor by scaling of bases. Therefore, our method has a smaller search space for low-rank tensors. This search space allows us to derive an efficient algorithm without a gradient method, which always outputs the globally optimal solution in the space.

## 4 Numerical Experiments

We empirically examined the efficiency and the effectiveness of LTR using synthetic and real-world datasets. We compared LTR with two existing non-negative low Tucker-rank approximation methods.

The first method is non-negative Tucker decomposition, which is the standard nonnegative tensor decomposition method [41] whose cost function is either the Least Squares (LS) error (NTD_LS) or the KL divergence (NTD_KL). The second method is sequential nonnegative Tucker decomposition (lraSNTD), which is known as the faster of the two methods [43]. Its cost function is the LS error; see Supplement for implementation details. We also obtained almost the same results as in Figure 3 with the KL reconstruction error (see Supplement).

**Results on Synthetic Data.** We created tensors with $d = 3$ or $d = 5$, where every $I_k = n$. We change $n$ to generate various sizes of tensors. Each element is sampled from the uniform continuous distribution on $[0, 1]$. To evaluate the efficiency, we measured the running time of each method. To evaluate the accuracy, we measured the LS reconstruction error, which is defined as the Frobenius norm between input and output tensors. Figure 3($\mathbf{a}$) shows the running time and the LS reconstruction error for randomly generated tensors with $d = 3$ and $n = 30$ with varying the target Tucker tensor rank. Figure 3($\mathbf{b}$) shows the running time and the LS reconstruction error for the target Tucker rank $(10, 10, 10)$ with varying the input tensor size $n$. These plots clearly show that our method is faster than other methods while keeping the competitive approximation accuracy.

**Results on Real Data.** We evaluated running time and the LS reconstruction error for two real-world datasets. 4DLFD is a $(9, 9, 512, 512, 3)$ tensor [18] and AttFace is a $(92, 112, 400)$ tensor [33]. AttFace is commonly used in tensor decomposition experiments [21, 22, 43]. For the 4DLFD dataset, we chose the target Tucker rank as (1,1,1,1,1), (2,2,2,2,1), (3,3,4,4,1), (3,3,5,5,1), (3,3,6,6,1), (3,3,7,7,1), (3,3,8,8,1), (3,3,16,16,1), (3,3,20,20,1), (3,3,40,40,1), (3,3,60,60,1), and (3,3,80,80,1). For the AttFace dataset, we chose (1,1,1), (3,3,3), (5,5,5), (10,10,10), (15,15,15), (20,20,20), (30,30,30), (40,40,40), (50,50,50), (60,60,60), (70,70,70), and (80,80,80). See dataset details in the Supplement. In both datasets, LTR is always faster than the comparison methods, as shown in Figure 3($\mathbf{c}$, $\mathbf{d}$), with competitive or better approximation accuracy in terms of the LS error.

As described above, the search space of LTR is smaller than that of NTD and lraSNTD. Nevertheless, our experiments show that the approximation accuracy of LTR is competitive with other methods. This means that NTD and lraSNTD do not effectively treat linear combinations of bases.

# 5   Conclusion

We have derived a new probabilistic perspective to rank-1 approximation for tensors using information geometry and shown that it can be viewed as mean-field approximation. Our new geometric understanding leads to a novel fast non-negative low Tucker-rank approximation method, called LTR, which does not use any gradient method. Our research will not only lead to applications of faster tensor decomposition, but can also be a milestone of the research of tensor decomposition to further development of interdisciplinary field around information geometry and the mean-field theory.

This study is a theoretical analysis of tensors and we believe that our theoretical discussion will not have negative societal impacts.

# Acknowledgement

This work was supported by JSPS KAKENHI Grant Number JP20J23179 (KG), JP21H03503 (MS), and JST, PRESTO Grant Number JPMJPR1855, Japan (MS).

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
