# A  Proof of Propositions

## A.1  Proof of Proposition 1

**Proposition 1** (Bingo and Tucker rank). *If there are $b_k$ bingos on mode-k, it holds that*

$$\mathrm{Rank}(\mathcal{P}^{(k)}) \le I_k - b_k.$$

*Proof.*  If there is a bingo on mode-$k$, the $m$-th row of the mode-$k$ expansion of $\mathcal{P}$ is a constant multiple of the $(m-1)$-th row, where $m$ is a number determined by the bingo position. Indeed,

$$
\begin{aligned}
\frac{\mathcal{P}^{(k)}_{m,j}}{\mathcal{P}^{(k)}_{m-1,j}} &= \frac{\mathcal{P}_{i_1,\ldots,i_{k-1},m,i_{k+1},\ldots,i_d}}{\mathcal{P}_{i_1,\ldots,i_{k-1},m-1,i_{k+1},\ldots,i_d}} \\
&= \frac{\exp\left(\sum_{i'_1=1}^{i_1}\cdots\sum_{i'_{k-1}=1}^{i_{k-1}}\sum_{i'_k=1}^{m}\sum_{i'_{k+1}=1}^{i_{k+1}}\cdots\sum_{i'_d=1}^{i_d}\theta_{i'_1,\ldots,i'_d}\right)}{\exp\left(\sum_{i'_1=1}^{i_1}\cdots\sum_{i'_{k-1}=1}^{i_{k-1}}\sum_{i'_k=1}^{m-1}\sum_{i'_{k+1}=1}^{i_{k+1}}\cdots\sum_{i'_d=1}^{i_d}\theta_{i'_1,\ldots,i'_d}\right)} \\
&= \exp\left(\sum_{i'_1=1}^{i_1}\cdots\sum_{i'_{k-1}=1}^{i_{k-1}}\sum_{i'_{k+1}=1}^{i_{k+1}}\cdots\sum_{i'_d=1}^{i_d}\theta_{i'_1,\ldots,i'_{k-1},m,i'_{k+1},\ldots,i'_d}\right) \\
&= \exp\left(\theta_{1,\ldots,1,m,1,\ldots,1}\right)
\end{aligned}
$$

is just a constant that does not depend on $j$. When a row is a constant multiple of another row, the rank of the matrix is reduced by a maximum of one, which means $\mathrm{Rank}(\mathcal{P}^{(k)}) \le I_k - 1$. In the same way, if there are $b_k$ bingos, then $b_k$ rows are constant multiple of the other rows, which means $\mathrm{Rank}(\mathcal{P}^{(k)}) \le I_k - b_k$. ☐

## A.2  Proof of Proposition 2

**Proposition 2** (rank-1 condition on $\theta$). *For any positive tensor $\overline{\mathcal{P}}$, $\mathrm{rank}(\overline{\mathcal{P}}) = 1$ if and only if its all many-body $\overline{\theta}$ parameters are $0$.*

*Proof.*  First, we show that $\mathrm{rank}(\overline{\mathcal{P}}) = 1$ implies all many-body $\theta$-parameters are $0$. From the assumption of $\mathrm{rank}(\overline{\mathcal{P}}) = 1$, the $m$-th row of the mode-$k$ expansion of $\overline{\mathcal{P}}$ have to be a constant multiple of the $(m-1)$-th row for all $m = \{2,\ldots,I_k\}$ and $k \in [d]$. That is,

$$
\frac{\overline{\mathcal{P}}^{(k)}_{m,j}}{\overline{\mathcal{P}}^{(k)}_{m-1,j}} = \frac{\overline{\mathcal{P}}_{i_1,\ldots,i_{k-1},m,i_{k+1},\ldots,i_d}}{\overline{\mathcal{P}}_{i_1,\ldots,i_{k-1},m-1,i_{k+1},\ldots,i_d}} = \exp\left(\sum_{i'_1=1}^{i_1}\cdots\sum_{i'_{k-1}=1}^{i_{k-1}}\sum_{i'_{k+1}=1}^{i_{k+1}}\cdots\sum_{i'_d=1}^{i_d}\theta_{i'_1,\ldots,i'_{k-1},m,i'_{k+1},\ldots,i'_d}\right)
$$

can depend on only $m$. If any many-body parameter $\theta_{i'_1,\ldots,m,\ldots i'_d}$ is not $0$ for $i'_1 + \cdots + i'_{k-1} + i'_{k+1} + \cdots + i'_d \ne d-1$, the left side of the above equation depends on indices other than $m$. For example, if a many-body parameter $\theta_{2,1,\ldots,1,m,1,\ldots 1}$ is not $0$, the equation depends on the value of $i_1$. Therefore, all many-body parameters of rank-1 tensor are $0$.

Next, we show that $\mathrm{rank}(\overline{\mathcal{P}}) = 1$ if all many-body $\theta$-parameters are $0$. If all many-body $\theta$-parameters are $0$, we have

$$
\overline{\mathcal{P}}_{i_1,\ldots,i_d} = \exp\left(\theta_{1,1,\ldots,1}\right)\prod_{k=1}^{d}\exp\left(\sum_{i'_k=2}^{i_k}\theta^{(k)}_{i'_k}\right).
$$

Then we can represent the tensor $\overline{\mathcal{P}}$ as the outer products of $d$ vectors $\boldsymbol{s}^{(1)} \in \mathbb{R}^{I_1}, \boldsymbol{s}^{(2)} \in \mathbb{R}^{I_2}, \ldots, \boldsymbol{s}^{(d)} \in \mathbb{R}^{I_d}$, whose elements are described as

$$
s^{(k)}_{i_k} = \exp\left(\frac{\theta_{1,\ldots,1}}{d}\right)\exp\left(\sum_{i'_k=2}^{i_k}\theta^{(k)}_{i'_k}\right)
$$

for each $k \in [d]$. Thus, $\mathrm{rank}(\overline{\mathcal{P}}) = 1$ followed by the definition of the tensor rank.

☐

## A.3 Proof of Proposition 3

The following proposition is related with the second paragraph in Section 3.4. We have succeeded in describing the rank-1 condition using $\eta$-parameter as well as on $\theta$-parameter.

**Proposition 3** (rank-1 condition as $\eta$ form). *For any positive $d$th-order tensor $\overline{\mathcal{P}} \in \mathbb{R}_{>0}^{I_1 \times \cdots \times I_d}$, $\mathrm{rank}(\overline{\mathcal{P}}) = 1$ if and only if its all many-body $\eta$ parameters are factorizable as*

$$\overline{\eta}_{i_1,\ldots,i_d} = \prod_{k=1}^{d} \overline{\eta}_{i_k}^{(k)}. \tag{14}$$

*Proof.* First, we show that all many-body $\eta$-parameters are factorizable if $\mathrm{rank}(\overline{\mathcal{P}}) = 1$. Since we can decompose a rank-1 tensor as a product of normalized independent distributions $\boldsymbol{s}^{(k)} \in \mathbb{R}^{I_k}$ as shown in Section 3.4, we can decompose many-body $\eta$ parameters of $\overline{\mathcal{P}}$ as follows:

$$
\begin{aligned}
\overline{\eta}_{i_1,\ldots,i_d} &= \sum_{i'_1=i_1}^{I_1} \cdots \sum_{i'_d=i_d}^{I_d} \overline{\mathcal{P}}_{i'_1,\ldots,i'_d} \\
&= \sum_{i'_1=i_1}^{I_1} \cdots \sum_{i'_d=i_d}^{I_d} \left( s_{i'_1}^{(1)} s_{i'_2}^{(2)} \cdots s_{i'_d}^{(d)} \right) \\
&= \left( \sum_{i'_1=i_1}^{I_1} s_{i'_1}^{(1)} \right) \left( \sum_{i'_2=i_2}^{I_2} s_{i'_2}^{(2)} \right) \cdots \left( \sum_{i'_d=i_d}^{I_d} s_{i'_d}^{(d)} \right) \\
&= \prod_{k=1}^{d} \left( \sum_{i'_k=i_k}^{I_k} s_{i'_k}^{(k)} \right) \\
&= \prod_{k=1}^{d} \left( \sum_{i'_k=i_k}^{I_k} s_{i'_k}^{(k)} \sum_{i'_1=1}^{I_1} s_{i'_1}^{(1)} \sum_{i'_2=1}^{I_2} s_{i'_2}^{(2)} \cdots \sum_{i'_d=1}^{I_d} s_{i'_d}^{(d)} \right) \\
&= \prod_{k=1}^{d} \left( \overline{\eta}_{i_k}^{(k)} \sum_{i'_k=1}^{I_k} s_{i'_k}^{(k)} \right) \\
&= \prod_{k=1}^{d} \overline{\eta}_{i_k}^{(k)},
\end{aligned}
$$

where we use the normalization condition

$$\sum_{i'_k=1}^{I_k} s_{i'_k}^{(k)} = 1$$

for each $k \in [d]$.

Next, we show the opposite direction. If all many-body $\eta$-parameters are factorizable, it follows that

$$
\begin{aligned}
\overline{\mathcal{P}}_{i_1,\ldots,i_d} &= \sum_{(i'_1,\ldots,i'_d) \in \Omega_d} \left( \mu_{i_1,\ldots,i_d}^{i'_1,\ldots,i'_d} \prod_{k=1}^{d} \overline{\eta}_{i'_k}^{(k)} \right) \\
&= \sum_{(i'_1 \ldots i'_d) \in \Omega_d} \left( \prod_{k=1}^{d} \mu_{i_k}^{i'_k} \overline{\eta}_{i'_k}^{(k)} \right) \\
&= \prod_{k=1}^{d} \left( \overline{\eta}_{i_k}^{(k)} - \overline{\eta}_{i_k+1}^{(k)} \right) \\
&\equiv \prod_{k=1}^{d} s_{j_k}^{(k)}.
\end{aligned}
$$

Thus, $\mathrm{rank}(\overline{\mathcal{P}}) = 1$ holds by the definition of the tensor rank. $\qquad\square$

## A.4 Proof of Proposition 4

The following proposition is related to the second paragraph in Section 3.4. The factorizability of $\eta$-parameter of rank-1 tensor and bingo rule reproduces the closed formula of the best rank-1 approximation minimizing KL divergence [3].

**Proposition 4** ($m$-projection onto factorizable subspace). *For any positive tensor $\mathcal{P} \in \mathbb{R}_{>0}^{I_1 \times \cdots \times I_d}$, its $m$-projection onto the rank-1 space is given as*

$$\overline{\mathcal{P}}_{i_1,\ldots,i_d} = \prod_{k=1}^{d}\left(\sum_{i'_1=1}^{I_1}\cdots\sum_{i'_{k-1}=1}^{I_{k-1}}\sum_{i'_{k+1}=1}^{I_{k+1}}\cdots\sum_{i'_d=1}^{I_d}\mathcal{P}_{i'_1,\ldots,i'_{k-1},i_k,i'_{k+1},\ldots,i_d}\right). \qquad (15)$$

*Proof.*

$$\overline{\mathcal{P}}_{i_1,\ldots,i_d} = \sum_{(i'_1\ldots i'_d)\in\Omega_d} \mu_{i_1\ldots i_d}^{i'_1,\ldots,i'_d}\overline{\eta}_{i'_1,\ldots,i'_d}$$

$$\overset{(14)}{=} \sum_{(i'_1,\ldots,i'_d)\in\Omega_d}\left(\mu_{i_1,\ldots,i_d}^{i'_1,\ldots,i'_d}\prod_{k=1}^{d}\overline{\eta}_{i'_k}^{(k)}\right)$$

$$\overset{(6)}{=} \sum_{(i'_1\ldots i'_d)\in\Omega_d}\left(\mu_{i_1,\ldots,i_d}^{i'_1,\ldots,i'_d}\prod_{k=1}^{d}\eta_{i'_k}^{(k)}\right)$$

$$= \sum_{(i'_1,\ldots,i'_d)\in\Omega_d}\left(\prod_{k=1}^{d}\mu_{i_k}^{i'_k}\eta_{i'_k}^{(k)}\right)$$

$$= \prod_{k=1}^{d}\left(\eta_{i_k}^{(k)}-\eta_{i_k+1}^{(k)}\right)$$

$$\overset{(4)}{=} \prod_{k=1}^{d}\left(\sum_{i'_1=1}^{I_1}\cdots\sum_{i'_{k-1}=1}^{I_{k-1}}\sum_{i'_{k+1}=1}^{I_{k+1}}\cdots\sum_{i'_d=1}^{I_d}\mathcal{P}_{i'_1,\ldots,i'_{k-1},i_k,i'_{k+1},\ldots,i_d}\right).$$

$\qquad\square$

Since the $m$-projection minimizes the KL divergence, it is guaranteed that $\overline{\mathcal{P}}$ obtained by Equation (15) minimizes the KL divergence from $\mathcal{P}$ within the set of rank-1 tensors. If a given tensor is not normalized, we need to divide the right-hand side of Equation (15) by the $d-1$-th power sum of all entries of the tensor in order to match the scales of the input and the output tensors. To summarize, the output of LTR $\overline{\mathcal{P}}$ is the best rank-1 approximation that always minimizes KL divergence,

$$\overline{\mathcal{P}} = \operatorname*{argmin}_{\hat{\mathcal{P}},\mathrm{rank}(\hat{\mathcal{P}})=1} D_{\mathrm{KL}}(\mathcal{P};\hat{\mathcal{P}}),$$

where the generalized KL divergence is defined as

$$D_{\mathrm{KL}}(\mathcal{P};\overline{\mathcal{P}}) = \sum_{i_1=1}^{I_1}\cdots\sum_{i_d=1}^{I_d}\left\{\mathcal{P}_{i_1\ldots i_d}\log\frac{\mathcal{P}_{i_1,\ldots,i_d}}{\overline{\mathcal{P}}_{i_1,\ldots,i_d}} - \mathcal{P}_{i_1,\ldots,i_d} + \overline{\mathcal{P}}_{i_1,\ldots,i_d}\right\}.$$

The generalized KL divergence for positive tensors is an extension of generalized KL divergence for non-negative matrices in [5], which enables us to treat non-normalized tensors.

## A.5 Proof of Proposition 5

The following discussion is related to the third paragraph in Section 3.4.

In the typical Boltzmann machine, which is defined as $p(\boldsymbol{x}) = \exp(\sum_i b_i x_i + \sum_{ij} w_{ij} x_i x_j)$ for a bias parameter $\boldsymbol{b} = (b)_i \in \mathbb{R}^n$, an interaction parameter $\boldsymbol{W} = (w_{ij}) \in \mathbb{R}^{n\times n}$, and a binary random

variable vector $x \in \{0, 1\}^n$, mean-field approximation is a projection onto the special manifold where $b_i = \log \frac{\eta_i}{1-\eta_i}$ holds for $\eta_i = \mathbb{E}_p[x_i]$.

In the rank-1 space, we show that $\theta$-parameters can be easily computed from $\eta$-parameters, as discussed in the Boltzmann machines in the following proposition; this supports our claim that rank-1 approximation can be regarded as mean-field approximation.

**Proposition 5.** *For any positive $d$th-order rank-1 tensor $\overline{\mathcal{P}} \in \mathbb{R}_{>0}^{I_1 \times \cdots \times I_d}$, its one-body $\eta$-parameters and one-body $\theta$-parameters satisfy the following equations*

$$\overline{\theta}_j^{(k)} = \log \left( \frac{\overline{\eta}_j^{(k)} - \overline{\eta}_{j+1}^{(k)}}{\overline{\eta}_{j-1}^{(k)} - \overline{\eta}_j^{(k)}} \right), \qquad \overline{\eta}_j^{(k)} = \frac{\sum_{i_k=j}^{I_k} \exp \sum_{i_k'=2}^{i_k} \overline{\theta}_{i_k'}^{(k)}}{1 + \sum_{i_k=2}^{I_k} \exp \left( \sum_{i_k'=2}^{i_k} \overline{\theta}_{i_k'}^{(k)} \right)},$$

*where we assume $\overline{\eta}_0^{(k)} = \overline{\eta}_{I_k+1}^{(k)} = 0$.*

*Proof.* As shown in Theorem 2 in Sugiyama et al. [9], the relation between $\theta$ and $\eta$ is obtained by the differentiation of Helmholtz's free energy $\psi(\theta)$, which is defined as the sign inverse normalization factor. For the rank-1 tensor $\overline{\mathcal{P}}$, Helmholtz's free energy $\psi(\theta)$ is given as

$$\psi(\overline{\theta}) = \log \prod_{k=1}^{d} \left( 1 + \sum_{i_k=2}^{I_k} \exp \left( \sum_{i_k'=2}^{i_k} \overline{\theta}_{i_k'}^{(k)} \right) \right).$$

We obtain the expectation parameters $\eta$ by the differentiation of Helmholtz's free energy $\psi(\theta)$ by $\theta$, given as

$$\overline{\eta}_j^{(k)} = \frac{\partial}{\partial \overline{\theta}_j^{(k)}} \psi(\overline{\theta}) = \frac{\sum_{i_k=j}^{I_k} \exp \sum_{i_k'=2}^{i_k} \overline{\theta}_{i_k'}^{(k)}}{1 + \sum_{i_k=2}^{I_k} \exp \left( \sum_{i_k'=2}^{i_k} \overline{\theta}_{i_k'}^{(k)} \right)}.$$

By solving the above equation inverse, we obtain

$$\overline{\theta}_j^{(k)} = \log \left( \frac{\overline{\eta}_j^{(k)} - \overline{\eta}_{j+1}^{(k)}}{\overline{\eta}_{j-1}^{(k)} - \overline{\eta}_j^{(k)}} \right).$$

$\square$

## A.6 Proof of Proposition 6

The following proposition is related to the third paragraph in Section 3.5.

**Proposition 6.** *Let $\theta$ denote canonical parameters of given tensor $\mathcal{P} \in \mathbb{R}^{I_1 \times \cdots \times I_d}$ and $\overline{\theta}$ denote canonical parameters of $\overline{\mathcal{P}}$ which is the best rank-1 approximation that minimizes KL divergence from $\mathcal{P}$. If some one-body canonical parameter $\theta_{i_j}^{(j)} = 0$ for some $i_j \in [I_j]$, its values after the best rank-1 approximation $\overline{\theta}_{i_j}^{(j)}$ remain 0.*

*Proof.* When $\theta_{i_j}^{(j)} = 0$, it holds that

$$\frac{\mathcal{P}_{1,\ldots,1,i_j,1,\ldots,1}}{\mathcal{P}_{1,\ldots,1,i_j-1,1,\ldots,1}} = \exp \left( \theta_{i_j}^{(j)} \right) = 1.$$

By using the closed formula of the best rank-1 approximation (15), we obtain

$$\overline{\mathcal{P}}_{1,\ldots,1,i_j,1,\ldots,1} = \prod_{k=1}^{d} \left( \sum_{i_1'=1}^{I_1} \cdots \sum_{i_{k-1}'=1}^{I_{k-1}} \sum_{i_{k+1}'=1}^{I_{k+1}} \cdots \sum_{i_d'=1}^{I_d} \mathcal{P}_{i_1',\ldots,i_{k-1}',1,i_{k+1}',\ldots,i_j,\ldots,i_d'} \right)$$

$$= \prod_{k=1}^{d} \left( \sum_{i_1'=1}^{I_1} \cdots \sum_{i_{k-1}'=1}^{I_{k-1}} \sum_{i_{k+1}'=1}^{I_{k+1}} \cdots \sum_{i_d'=1}^{I_d} \mathcal{P}_{i_1',\ldots,i_{k-1}',1,i_{k+1}',\ldots,i_j-1,\ldots,i_d'} \right)$$

$$= \overline{\mathcal{P}}_{1,\ldots,1,i_j-1,1,\ldots,1}.$$

It follows that

$$\frac{\overline{\mathcal{P}}_{1,\ldots,1,i_j,1,\ldots,1}}{\overline{\mathcal{P}}_{1,\ldots,1,i_j-1,1,\ldots,1}} = \exp\left(\overline{\theta}_{i_j}^{(j)}\right) = 1.$$

Finally, we obtain $\overline{\theta}_{i_j}^{(j)} = 0$. $\qquad\square$

### A.7 Proof of Theorem 1

**Theorem 1** (LTR). *For a positive tensor $\mathcal{P} \in \mathbb{R}_{>0}^{I_1 \times \cdots \times I_d}$, the $m$-projection destination onto the bingo space $\mathcal{B} = \mathcal{B}^{(1)} \cap \cdots \cap \mathcal{B}^{(d)}$ is given as iterative application of $m$-projection $d$ times, starting from $\mathcal{P}$ onto subspace $\mathcal{B}^{(1)}$, and then from there onto subspace $\mathcal{B}^{(2)}$, ..., and finally onto subspace $\mathcal{B}^{(d)}$.*

*Proof.* Let $\eta$ denote $\eta$-parameters of $\mathcal{P}$. Let $\Omega_{\mathcal{B}^{(k)}}$ be the set of bingo indices for $\mathcal{B}^{(k)}$:

$$\mathcal{B}^{(k)} = \left\{ \mathcal{P} \mid \theta_{i_1,\ldots,i_d} = 0 \text{ for } (i_1,\ldots,i_d) \in \Omega_{\mathcal{B}^{(k)}} \right\}.$$

In the first $m$-projection onto $\mathcal{B}^{(1)}$ from input $\mathcal{P}$, given us $\mathcal{P}_{(1)}$ whose parameters satisfy

$$\begin{aligned}
\tilde{\theta}_{i_1,\ldots,i_d} &= 0 && \text{if } (i_1,\ldots,i_d) \in \Omega_{\mathcal{B}^{(1)}}, \\
\tilde{\eta}_{i_1,\ldots,i_d} &= \eta_{i_1,\ldots,i_d} && \text{otherwise.}
\end{aligned}$$

The $\theta$-condition comes from the definition of the bingo space $\mathcal{B}^{(1)}$ and the $\eta$-condition comes from the conservation low of $\eta$-parameters in Equation (6). The second $m$-projection onto $\mathcal{B}^{(2)}$ from $\mathcal{P}_{(1)}$, we get $\mathcal{P}_{(1,2)}$ whose parameters satisfy

$$\begin{aligned}
\tilde{\theta}_{i_1,\ldots,i_d} &= 0 && \text{if } (i_1,\ldots,i_d) \in \Omega_{\mathcal{B}^{(1)}} \cup \Omega_{\mathcal{B}^{(2)}}, \\
\tilde{\eta}_{i_1,\ldots,i_d} &= \eta_{i_1,\ldots,i_d} && \text{otherwise.}
\end{aligned}$$

Proposition 6 ensures the above $\theta$ condition. The conservation low of $\eta$-parameters ensures the above $\eta$ condition. Similarly, in the final $m$-projection onto $\mathcal{B}^{(d)}$ from input $\mathcal{P}_{(1,\ldots,d-1)}$, we get $\mathcal{P}_{(1,\ldots,d-1,d)}$ whose parameters satisfy

$$\begin{aligned}
\tilde{\theta}_{i_1,\ldots,i_d} &= 0 && \text{if } (i_1,\ldots,i_d) \in \Omega_{\mathcal{B}^{(1)}} \cup \Omega_{\mathcal{B}^{(2)}} \cup \cdots \cup \Omega_{\mathcal{B}^{(d)}}, \\
\tilde{\eta}_{i_1,\ldots,i_d} &= \eta_{i_1,\ldots,i_d} && \text{otherwise.}
\end{aligned}$$

The distribution satisfying these two conditions is the $m$-projection from $\mathcal{P}$ to $\mathcal{B}$. $\qquad\square$

## B  Theoretical Remarks

**Invariance of the summation in each axial direction**  The definition of $\eta$ in Equation (4) suggests that one-body $\eta$-parameters are related to the summation of elements of a tensor in each axial direction. The $i_k$-th summation in the $k$-th axis is given by

$$\sum_{i_1=1}^{I_1} \cdots \sum_{i_{k-1}=1}^{I_{k-1}} \sum_{i_{k+1}=1}^{I_{k+1}} \cdots \sum_{i_d=1}^{I_d} \mathcal{P}_{i_1,\ldots,i_d} = \eta_{i_k}^{(k)} - \eta_{i_{k+1}}^{(k)}.$$

Since the one-body $\eta$-parameters do not change by the $m$-projection, it can be immediately proved that the best rank-1 approximation of a positive tensor in the sense of the KL divergence does not change the sum in each axial direction of the input tensor. Our information geometric insight leads to the fact that the conservation law of sums essentially comes from constant one-body $\eta$-parameters during $m$-projection. This property is a natural extension of the property, such that row sums and column sums are preserved in NMF, which minimizes the KL divergence [1] to tensors. Since the rank-1 reduction preserves the sum in each axial direction of the input tensor, LTR for general Tucker rank reduction also preserves it.

## C  Experiment Setup

All evaluation code are attached in the supplementary material.

**Implementation Details**   All methods were implemented in `Julia` 1.6 with `TensorToolbox`[1] library [7], hence runtime comparison is fair. We implemented lraSNTD referring to the original papers [10]. We used the `TensorLy` implementation [4] for NTDs. Experiments were conducted on CentOS 6.10 with a single core of 2.2 GHz Intel Xeon CPU E7-8880 v4 and 3TB of memory. We use default values of hyper parameters of `tensorly` [4] for NTD. We used default values of the hyper parameters of `sklearn` [6] for the NMF module in lraSNTD.

**Dataset Details**   We describe the details of each dataset in the following.   4DLFD is a $(9, 9, 512, 512, 3)$ tensor, which is produced by 4D Light Field Dataset described in [2]. Its license is Creative Commons Attribution-Non-Commercial-ShareAlike 4.0 International License. We used dino images and their depth and disparity map in training scenes and concatenated them to produce a tensor. AttFace is a $(92, 112, 400)$ tensor that is produced by the entire data in The Database of Faces (AT&T) [8], which includes 400 grey-scale face photos. The size of each image is $(92, 112)$. AttFace is public on Kaggle but the license is not specified.

# D   Tensor Operations

## D.1   mode-k expansion

The mode-$k$ expansion of a tensor $\mathcal{P} \in \mathbb{R}^{I_1 \times \cdots \times I_d}$ is an operation that focuses on the $k$th axis of the tensor $\mathcal{P}$ and converts $\mathcal{P} \in \mathbb{R}^{I_1 \times \cdots \times I_d}$ into a matrix $\mathcal{P}^{(k)} \in \mathbb{R}^{I_k \times \prod_{m=1(m \neq k)}^{d} I_m}$. The relation between tensor $\mathcal{P}$ and its mode-$k$ expansion $\mathcal{P}^{(k)}$ is given as,

$$\left( \mathcal{P}^{(k)} \right)_{i_k, j} = \mathcal{P}_{i_1, \dots, i_d},$$

$$j = 1 + \sum_{l=1, (l \neq k)}^{d} (i_l - 1) J_l,$$

$$J_l = \prod_{m=1, (m \neq k)}^{l-1} I_m.$$

## D.2   Kronecker product for vectors

Given $d$ vectors $\boldsymbol{a}^{(1)} \in \mathbb{R}^{I_1}$, $\boldsymbol{a}^{(2)} \in \mathbb{R}^{I_2}$, ..., $\boldsymbol{a}^{(d)} \in \mathbb{R}^{I_d}$, the Kronecker product $\mathcal{P}$ of these $d$ vectors, written as

$$\mathcal{P} = \boldsymbol{a}^{(1)} \otimes \boldsymbol{a}^{(2)} \otimes \cdots \otimes \boldsymbol{a}^{(d)},$$

is a tensor in $\mathbb{R}^{I_1 \times \cdots \times I_d}$, where each element of $\mathcal{P}$ is given as

$$\mathcal{P}_{i_1, \dots, i_d} = a_{i_1}^{(1)} a_{i_2}^{(2)} \dots a_{i_d}^{(d)},$$

where $a_i^{(k)}$ is an $i$th element of a vector $\boldsymbol{a}^{(k)}$.

# E   Experimental results with KL divergence

The cost function of LTR is the KL divergence from the input tensor to the low-rank tensor. Our experimental results in Figure 4 show that LTR also has better or competitive accuracy of approximation in terms of the KL divergence.

---

[1]MIT Expat License

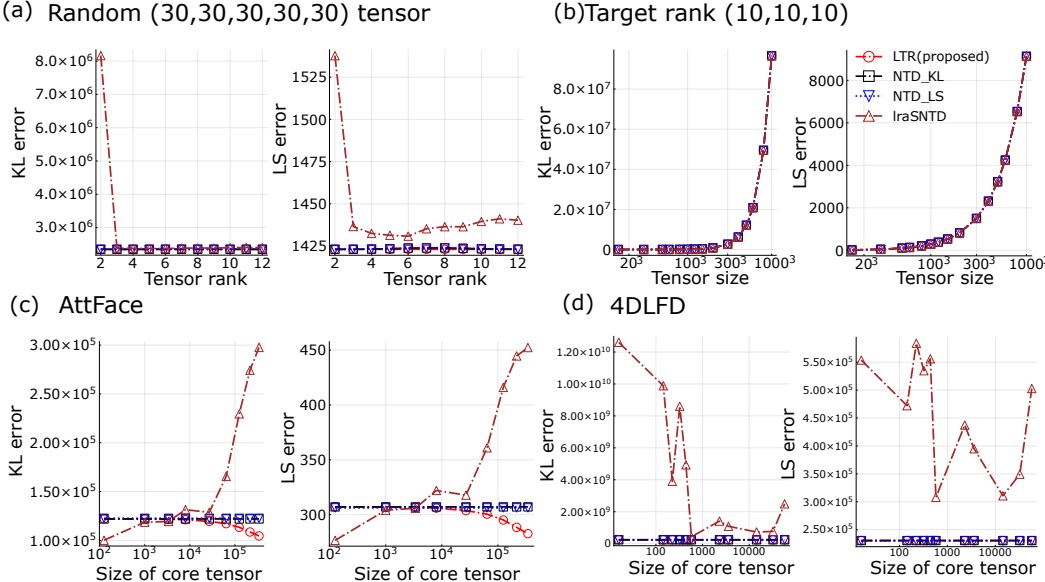

Figure 4: Experimental results for synthetic (a, b) and real-world (c, d) datasets. The left-hand panels are KL reconstruction error and the right-hand panels are LS reconstruction error. (a) The horizontal axis is $r$ for target tensor rank $(r, r, r, r, r)$. (b) The horizontal axis is $n^3$ for input $(n, n, n)$ tensor.