# OpenReview forum: "Fast Tucker Rank Reduction for Non-Negative Tensors Using Mean-Field Approximation"
_NeurIPS.cc/2021/Conference — NeurIPS 2021 Poster_

### Official Review · Reviewer_8bMP · 2021-07-13

**Rating:** 6
**Confidence:** 3

**Summary:**

This paper derives a tensor decomposition method using mean-field approximation and information geometry. The main claim of this paper is the speed of computation compared to some basic baseline methods. An interesting direction but there are several directions that the paper can improve further.

**Limitations And Societal Impact:**

Only the advantage of the speed of decomposition is discussed. It is useful if reconstruction accuracy is also improved.

Another limitation is the restriction to Tucker decomposition, since recently developed methods such as tensor trains can be more feasible to use with higher-order tensors.

**Main Review:**

This paper is technically interesting and can be a useful addition to the tensor decomposition method. However, this new method does not show any additional benefits other than the gain in speed.

Experiments are not convincing. First of all, I feel that the dimensions for synthetic data are small. I would like to see the speed for much larger tensors. It would be informative if the Tucker ranks of tensor have different values, rather than considering equal r. I believe that it will be challenging to identify the Tucker rank in cases where there is not equal r.
From the given plots it is difficult to identify differences in accuracy between the proposed method and the NTD. Further, it is hard to understand what is meant by the size of the core tensor, can the authors explain it (and how it is decided)?
Can authors include simulations with larger tensors that with different Tucker ranks on each mode?   Can the authors provide more details on the prediction accuracy of tensor decomposition?

It would be useful to know if the proposed methods mean-field approximation can be extended to other tensor decomposition methods such as tensor networks. Tensor networks such as tensor trains have shown to be more computationally robust for higher-order tensors.

The paper is written well. I am not highly familiar with all the methods used discussed in the paper, however, the paper seems to be technically correct. The plots in the experiments section should be large enough to see lines clearly. Can the authors give core-computational steps in algorithmic format for easy understanding?

**Time Spent Reviewing:**

1.5 hour

---

> ### Author Response · Authors · 2021-08-09
> **Authors’ response to Reviewer 8bMP**
>
> > The dimensions for synthetic data are small.
>
> We believe that both synthetic datasets are large enough to evaluate the efficiency of LTR. In Figure 3(a), the tensor size is (30,30,30,30,30), thus the number of elements becomes 24,300,000. In Figure 3(b), we have examined tensors whose size is up to (1000,1000,1000), where the number of elements is 1000^3, that is, one billion.
>
> > It would be informative if the Tucker ranks of tensor have different values … Can authors include simulations with larger tensors that with different Tucker ranks on each mode?
>
> We already have such results in Figure 3(d), where we consider different target Tucker ranks on each mode. As you can see in `config.jl` for 4DLFD dataset in our supplement, we chose the target tucker rank as [1,1,1,1,1], [2,2,2,2,1], [3,3,4,4,1], [3,3,5,5,1], [3,3,6,6,1], [3,3,7,7,1], [3,3,8,8,1], [3,3,16,16,1], [3,3,20,20,1], [3,3,40,40,1], [3,3,60,60,1], and [3,3,80,80,1]. We will explicitly explain this point in the final version.
>
> > From  the given plots it is difficult to identify differences in accuracy between the proposed method and the NTD. The plots in the experiments section should be large enough to see lines clearly.
>
> We believe that each line in Figure 3 can be clearly seen as we have used different marks (circle, square, triangle, and inverse triangle), line types, and colors, and it is clear enough to support our claim that the approximation accuracy of LTD is comparable to NTD. For completeness, we provide the raw numbers of our experiments on real-world datasets in the following.
>
> 4DLFD	LS error
>
> | size of core | lraSNTD     | NTD\_LS     | NTD\_KL     | LTR         |
> | ------------ | ----------- | ----------- | ----------- | ----------- |
> | 1            | 2.30245E+05 | 2.30242E+05 | 2.30588E+05 | 2.30588E+05 |
> | 16           | 5.53739E+05 | 2.30242E+05 | 2.30588E+05 | 2.30588E+05 |
> | 144          | 4.72100E+05 | 2.30242E+05 | 2.30588E+05 | 2.30588E+05 |
> | 225          | 5.83872E+05 | 2.30242E+05 | 2.30588E+05 | 2.30588E+05 |
> | 324          | 5.34740E+05 | 2.30242E+05 | 2.30588E+05 | 2.30588E+05 |
> | 441          | 5.56001E+05 | 2.30242E+05 | 2.30588E+05 | 2.30588E+05 |
> | 576          | 3.07946E+05 | 2.30242E+05 | 2.30588E+05 | 2.30588E+05 |
> | 2304         | 4.37373E+05 | 2.30242E+05 | 2.30588E+05 | 2.30588E+05 |
> | 3600         | 3.94962E+05 | 2.30242E+05 | 2.30588E+05 | 2.30588E+05 |
> | 14400        | 3.10697E+05 | 2.30242E+05 | 2.30588E+05 | 2.30588E+05 |
> | 32400        | 3.49382E+05 | 2.30242E+05 | 2.30588E+05 | 2.30588E+05 |
> | 57600        | 5.02847E+05 | 2.30242E+05 | 2.30588E+05 | 2.30588E+05 |
>
> AttFace	 LS error
>
> | size of core | lraSNTD | NTD\_LS | NTD\_KL | LTR     |
> | ------------ | ------- | ------- | ------- | ------- |
> | 1            | 306.232 | 306.215 | 307.111 | 307.111 |
> | 125          | 276.599 | 306.215 | 307.111 | 307.084 |
> | 1000         | 304.086 | 306.215 | 307.111 | 307.025 |
> | 3375         | 305.878 | 306.215 | 307.111 | 306.73  |
> | 8000         | 322.097 | 306.215 | 307.111 | 306.021 |
> | 27000        | 317.782 | 306.215 | 307.111 | 303.762 |
> | 64000        | 360.928 | 306.215 | 307.111 | 300.482 |
> | 125000       | 416.039 | 306.215 | 307.111 | 295.163 |
> | 216000       | 444.654 | 306.215 | 307.111 | 288.437 |
> | 343000       | 452.315 | 306.215 | 307.111 | 282.697 |
>
> > What is meant by the size of the core tensor, and how it is decided?
>
> The size of the core tensor means the number of elements of the core tensor; that is, $r_1 r_2 \cdots r_d$ for a tensor with the Tucker rank $(r_1, \ldots , r_d)$. In our experiment using the dataset 4DLFD in Figure 3(c), since its size is $(9,9,512,512,3)$ and the dimension of each mode varies a lot, the target tacker ranks (and the corresponding size of the core tensor) were decided as described in our previous answer. In Figure 4(d) with the dataset AttData, since its size is $(112,92,400)$ and the difference between modes is not so large, we set the same target Tucker rank across modes. We will add this explanation in the final version.
>
> > Can the authors provide more details on the prediction accuracy of tensor decomposition?    Only the advantage of the speed of decomposition is discussed. It is useful if reconstruction accuracy is also improved.
>
> We have two observations about reconstruction errors as follows.
>
> 1. When the target rank is 1, our method LTR always provides the best approximation which minimizes the KL divergence from the input tensor (see l196-197).
> 2. When the target rank is the same as the tensor size, it is guaranteed that the output of LTR is exactly the same as input since any replacement in Step 2 is not conducted. Thus, the reconstruction error becomes 0, which is not guaranteed for NTDs.
>
> Please note that, as we have shown in experiments, our method also shows competitive or better approximation performance than NTD.
>
> > It would be useful to know if the proposed methods mean-field approximation can be extended to other tensor decomposition methods such as tensor networks.
>
> Thank you for an interesting suggestion. We believe that our approach, which models a tensor using discrete partial order structure, has a flexibility to extend to other tensor tasks. Tensor train based decomposition is out of our scope and our method cannot be directly applied to in its current form. However, we think it might be possible to develop a tensor train decomposition based on our discussion in the following three steps.
> 1.	Design a poset appropriately.
> 2.	Describe the low-rank $\theta$ condition in the tensor train decomposition.
> 3.	Perform $m$-projection onto the subspace satisfying the condition specified in the above step.
>
> Although we need more consideration to determine whether the tensor train rank-1 decomposition can be regarded as a mean-field approximation, it is definitely our interesting future work, and this paper may give some insight into these challenges.
>
> > Can the authors give core-computational steps in algorithmic format for easy understanding?
>
> We will add the algorithmic format in the appendix in the final version. Here, we provide it as a markdown format. Note that we denote by $P_{a^{(k)}:b^{(k)}}$ the subtensor obtained by fixing the range of $k$th index to only from $a$ to $b$ (see l78-79).
>
> ---
>
> **Input:** tensor $P$, target Tucker rank $\mathbf{r}=(r_1,r_2,\dots,r_d)$
>
> **Output:** rank reduced tensor $Q$
>
> ---
>
> #### *LTR$\left(P, \mathbf{r} \right)$*:
> $(I_1,\dots,I_d) \leftarrow$ the size of each mode of the input tensor $P$
> **foreach** $k=1,\dots,d$ **do**
>     &emsp;Sample $r_k$ numbers from $\{2, 3, \ldots, I_k\}$ without replacement and put them into $c_2, c_3, \ldots, c_{r_k}$,
> &emsp;where we always assume that $c_1=1$ and $c_i < c_{i+1}$
> $Q \leftarrow P$
> **foreach** $l=1,\dots,d$ **do**
>    &emsp; **if** $c_l \neq c_{l+1}-1$ **then**
>       &emsp;&emsp;  Replace the subtensor $Q_{c_l^{(k)} :  c_{l+1}^{(k)}-1}$ of $Q$ by its rank-1 approximation as
>       &emsp;&emsp;  $Q_{c_l^{(k)} :  c_{l+1}^{(k)}-1} \leftarrow $ *BestRankOne* $\left( Q_{c_l^{(k)} :  c_{l+1}^{(k)}-1}\right)$
> **return** $Q$
>
> ---
>
> #### *BestRankOne$\left(P\right)$*:
> $(I_1,\dots,I_d) \leftarrow$ the size of each mode of the input tensor $P$
> **foreach** $k=1,\dots,d$ **do**
>     &emsp;**foreach** $i_k=1,\dots,I_k$ **do**
>         &emsp;&emsp;$s_{i_k}^{(k)} \leftarrow \sum_{i_1} \dots \sum_{i_{k-1}}\sum_{i_{k+1}}\dots\sum_{i_d} P_{i_1, \dots i_{k-1},i_k,i_{k+1},\dots,i_d}$
> $\lambda \leftarrow $ sum of all elements of $P$
> $ \overline P \leftarrow
>     \lambda {\boldsymbol s}^{(1)}
>     \otimes
>     {\boldsymbol s}^{(2)}
>     \otimes \dots \otimes
>     {\boldsymbol s}^{(d)} $
> **return** $\overline  P$
>
> ---

---

> > ### Comment · Reviewer_8bMP · 2021-08-29
> > **Reply**
> >
> > Thank you for the reply.
> >
> > I think inclusion of the algorithmic details is essential in the revised paper. Also, it would be helpful (not essential) to have some comments on how to extend the method to other decomposition method.
> >
> > I still wonder about the practical advantages of the proposed methods besides the gain in speed of computations?

---

> > > ### Author Response · Authors · 2021-08-30
> > > **Reply**
> > >
> > > Thank you for your comments.
> > >
> > > > I think the inclusion of the algorithmic details is essential in the revised paper.
> > >
> > > We agree with this point. We promise that we include the algorithmic format, which we have described in our initial response, in the appendix in the final version.
> > >
> > > > I still wonder about the practical advantages of the proposed methods besides the gain in speed of computations?
> > >
> > > We can solve our convex optimization (l.136-138) using the analytical formula of the best rank-1 approximation, hence we do not need a gradient method (l.34-38) unlike other methods. Thus, we do not have to tune learning rate, stopping criteria, or initialization, which is convenient to the user and is also a practical advantage of our method.
> > >
> > > In addition, we can include prior knowledge for better low-rank approximation (l.93-95). If some prior knowledge is available and we know that some part of an input tensor is less important than the other part in advance, we can choose such uninformative indices for making bingos. The resulting tensor is expected to achieve better reconstruction.

---

### Official Review · Reviewer_3L1X · 2021-07-14

**Rating:** 4
**Confidence:** 3

**Summary:**

This paper considers the problem of low-rank approximation for non-negative tensors. The authors propose a Tucker rank reduction algorithm based on information geometry, and provide a probabilistic perspective through mean-field approximation.

**Limitations And Societal Impact:**

Yes

**Main Review:**

As the paper is mainly focused on the theoretical analysis of low-rank tensor approximation, I think the main drawback is that there is no theoretical guarantee for both the approximation error and the computational complexity of the algorithm. The performance of the algorithm is only shown by numerical experiments. In addition, the proposed sufficient condition and the mean-field connection seem to be of only theoretical interest, and fail to provide insights and guidance for practical applications.


**Time Spent Reviewing:**

4

---

> ### Author Response · Authors · 2021-08-09
> **Authors’ response to Reviewer 3L1X**
>
> > There is no theoretical guarantee for the approximation error.
>
> Although we do not have an explicit bound of the approximation error in our method LTR, as we have mentioned in l286-288, we have a strong theoretical guarantee that LTR always provides the *best* low-rank approximation in the specified bingo space in terms of the KL divergence; that is, it always outputs the globally optimal tensor among any tensors that can be represented within the specified bingo space.
>
> > There is no theoretical guarantee for computational complexity.
>
> This argument is not correct. Our method LTR has the theoretical guarantee for the computational complexity as we have already mentioned in l92-93. We also would like to clarify the complexity of each step of LTR in the following. For an input tensor with the size of $(I_1, \ldots, I_d)$ and the target rank $(r_1, \ldots, r_d)$, Step 1 of LTR requires $O(r_1 + r_2 + \cdots + r_d)$ since all we need is to sample $r_k$ integers from ${1,2,\ldots, I_k}$ for each $k \in [d]$ using the Fisher-Yates method, which is trivial and we did not mention in our paper. Step 2 takes $O(r_1\cdots r_d I_1\cdots I_d)$ as we have mentioned in l92-93, which is the overall complexity of LTR.
>
> > The proposed sufficient condition and the mean-field connection seem to be of only theoretical interest, and fail to provide insights and guidance for practical application.
>
> We disagree with this comment. The tight connection between the sufficient condition, which we call bingo rule, and the mean-field approximation is indispensable to construct a practical low-rank approximation algorithm LTR. More precisely, we have shown that LTR always provides the best low-rank approximation in the selected bingo space. Therefore, the bingo rule is necessary to prove the optimality of LTR, which is of not only theoretical but practical interest.
>
> In addition, the concept of bingos could provide practical guidance for the user as there is a clear interpretation that the information on bingos will be lost in low-rank approximation by LTR. For example, as we have mentioned in l93-95, if prior knowledge is available and we know that some part of an input tensor is less important than the other part, we can choose such uninformative indices for making bingos. Then it is expected that the resulting tensor will have better reconstruction error.

---

### Official Review · Reviewer_TkCd · 2021-07-17

**Rating:** 7
**Confidence:** 2

**Summary:**

The paper considers the low Tucker-rank approximation problem, uses the mean-field theory to interpret rank-one approximation for tensors by treating tensors as probability distributions, and then develops a two-step Tucker-rank reduction algorithm which does not involve gradient descent. Theoretical analysis of the proposed algorithm is provided in the information geometry framework. In particular, a Bingo rule is introduced to relate Tucker rank and modal expansion of a tensor in another coordinate system, which paves the way to reformulate the Tucker rank reduction problem as an m-projection.

**Limitations And Societal Impact:**

The proposed framework works only for nonnegative tensors due to the exponential involved coordinate transform, which may limit the use in practice. Moreover, if the number of modes or some dimension is very large, the first step of LTR involves a random sampling which may waste some iterations and computation of (1) could be slow. Stopping criteria for LTR could be mentioned as well.

**Main Review:**

The proposed tensor-rank reduction method is interesting, and theoretical discussions from the perspective of mean-field theory has certain novelty. Numerical results on various types of data sets are supportive and convincing. This work is important for low-rank approximations in tensor recovery and could be insightful for other tensor rank reduction methods. Minor issues: In line 19-20, there are more tensor ranks other than CP-rank and Tucker-rank, e.g., tubal rank. In line 84, what does "the total sum of $\mathcal{P}$" mean? In line 94, what are "uninformative indices"? Maybe some examples could be referred. In the caption of Figure 1, "Bingo space" is not defined. In lines 211-212, how can the distribution independence assumption be guaranteed for generic tensors in the original form? Does the performance of the algorithm rely on this assumption? In Figure 3, are there any reasons for choosing the tensors with the same dimension across the modes?

**Time Spent Reviewing:**

1

---

> ### Author Response · Authors · 2021-08-09
> **Authors’ response to Reviewer TkCd**
>
> >  In line 19-20, there are more tensor ranks other than CP-rank and Tucker-rank, e.g., tubal rank.
>
> Thank you for your suggestion. We will mention and cite such other tensor ranks in the final version.
>
> > In line 84, what does "the total sum of P" mean?
>
> This means the sum of all elements of $\mathcal{P}$, that is, $\sum_{i_1, i_2, \ldots, i_d} \mathcal{P}_{i_1, i_2, \ldots, i_d}$. We will add this explanation in the final version.
>
> > In line 94, what are "uninformative indices"?
>
> Currently, we randomly choose bingo indices in Step 1 by the uniform distribution. If we know that some parts of an input tensor are less important than the other parts from prior knowledge, we can directly choose these indices for making bingos instead of random sampling to obtain a more accurate reconstructed tensor. We call these indices uninformative indices.
>
> > In the caption of Figure 1, "Bingo space" is not defined.
>
> Thank you for pointing this out. In the final version, we will insert a sentence, “We call the low-rank subspace $\mathcal{B}$ bingo space” in l162.
>
> > In lines 211-212, how can the distribution independence assumption be guaranteed for generic tensors in the original form? Does the performance of the algorithm rely on this assumption?
>
> Mean-field approximation is the task of approximating a joint distribution by a product of independent distributions, and we do not need to assume the distribution independence for input tensors. Even if the target distribution is not generated from independent distributions, it is well known that mean-field approximation can approximate it well, for example, in the Ising model. In our paper, we have shown that the rank-1 approximation of a tensor corresponds to approximation the original tensor by a product of independent distributions (l202-l210). If the original tensor is generated from a product of independent distributions, that is, the rank of the original tensor is already 1   , the mean-field approximation exactly represents the original tensor and the reconstruction error becomes zero, which does not hold in general.
>
> > In Figure 3, are there any reasons for choosing the tensors with the same dimension across the modes?
>
> We have evaluated the performance of our method LTR on tensors with the same dimension across the modes (Figure 3(a), (b)) and also tensors with different dimensions across the modes (Figure 3(c), (d)).
>
> > The proposed framework works only for nonnegative tensors due to the exponential involved coordinate transform, which may limit the use in practice.
>
> Thank you for your suggestion. We will clarify this point as the limitation of our work. Nevertheless, there are many situations in data analysis where it is reasonable to impose non-negative constraints such as images, videos, purchase history, and traffic data.
>
> > If the number of modes or some dimension is very large, the first step of LTR involves a random sampling which may waste some iterations and computation of (1) could be slow.
>
> This point is not our limitation as random sampling does not cause a loss of computational efficiency even for tensors with higher dimensions. The time complexity required to extract $r_j$ elements from an array with the length $I_j$ without replacement is $O(r_j)$ using the Fisher-Yates method. Thus the total complexity of Step 1 is $O(r_1 + \cdots + r_d)$, which is linear with respect to the number of dimensions. In addition, Step 2 requires $O(I_1\cdots l_d r_1 \cdots r_d)$ as mentioned in l93.
>
> > Stopping criteria for LTR could be mentioned as well.
>
> LTR is not an iterative method, hence it needs neither stopping criteria, learning rate, nor tuning of initialization. As you can see in the final paragraph on page1, this is also our advantage．

---

### Official Review · Reviewer_F6qZ · 2021-07-19

**Rating:** 7
**Confidence:** 3

**Summary:**

The paper presents a low-rank tensor approximation algorithm for non-negative tensors. The algorithm is based on two observations. First, rank-1 approximation for tensors can be viewed as a mean-field approximation by treating each tensor as a probability distribution. Second, the authors provide a sufficient condition for distribution parameters to reduce Tucker ranks of tensors and, interestingly, this sufficient condition can be achieved by iterative application of the mean-field approximation. Since the mean-field approximation is always given as a closed formula, a fast low-rank approximation algorithm can be formulated without using a gradient method. The authors empirically demonstrate on several datasets that the algorithm is faster than the existing non-negative Tucker rank reduction methods ( in terms of achieving comparable approximations of given tensors).

**Limitations And Societal Impact:**

The paper does not include any narrative of any negative societal impact. That said, I don't immediately see an immediate negative impact beyond the hypothetical possibility of tensor decompositions being used for unethical downstream tasks.

**Main Review:**

Pros
* Strong theoretical and algorithmic contribution with an original connection between tensor factorizations and mean-field approximation theory.
* The proposed algorithm is novel and interesting
* I'd rate the significance of the work as high as the paper draws on ideas from information geometry and physics. Hence the paper will likely be of interest to diverse sets of researchers.
* The paper is clearly and well written.

Cons
* The paper is a bit sparse on the experimental side but that's not necessarily a con given the algorithmic/theoretical contribution as mentioned above.


**Time Spent Reviewing:**

3

---

> ### Author Response · Authors · 2021-08-09
> **Authors’ response to Reviewer F6qZ**
>
> Thank you very much for your positive comments.

---

### Decision · Program_Chairs · 2021-09-27

**Decision:**

Accept (Poster)

**Comment:**

This paper gives a new way to do Tucker rank reduction for non-negative tensors. The new approach puts additional constraints on the form of the decomposition (characterized by "bingo-space" defined in the paper) and allows the algorithm to find the optimal solution under these constraints very efficiently. The approach is quite novel and has some reasonable guarantees. The authors should clarify these guarantees in detail in the revised version.